# Improving the EAR Index for Flexible Pavement and a Preliminary Definition of an Environmental Index (ECR) for Rigid Pavement

Cristina Tozzo [1,*], Davide Chiola [2], Matteo Pierani [1], Leonardo Urbano [1], Riccardo Ricci [2] and Stefano Susani [3]

1   TECNE S.p.A., 20126 Milan, Italy; matteo.pierani@tecneautostrade.it (M.P.);
    leonardo.urbano@tecneautostrade.it (L.U.)
2   Movyon S.p.A., 50013 Florence, Italy; davide.chiola@movyon.it (D.C.); riccardo.ricci@movyon.it (R.R.)
3   Amplia Infrastructures S.p.A., 00156 Rome, Italy; stefano.susani@autostrade.it
*   Correspondence: cristina.tozzo@tecneautostrade.it

**Abstract:** Nowadays, roadway infrastructures are designed in order to satisfy technical and economical requirements, as well as to guarantee advanced environmental performance. Focusing on that, this paper deals with an innovative procedure for the characterization of pavement materials, both asphalt and cement-bound mixtures. The methodology takes its cue from a previous study in which the so-called Environmental Asphalt Rating (EAR) was firstly introduced as a reference parameter for asphalt pavements to evaluate technical offers and for the assignment of scores, in terms of environmental impacts, during the tender phase. In this work, the EAR methodology is revised with a focus on the main variations and improvements related to the new version of the ISO standard. By applying the same approach to rigid or concrete pavements, a preliminary version of the Environmental Concrete Rating (ECR) is presented. For ECR, a correction is provided regarding functionality through a fatigue-related parameter and the surface characteristics related to the IRI value. Despite its strong applicability to the pavement sector, the strength of the proposed method is its ability to be fine-tuned to different fields by varying the associated performance coefficients.

**Keywords:** LCA; pavement materials; performance analysis; environmental assessment





## 1. Introduction

Nowadays, transportation infrastructures are designed to satisfy technical and economical requirements as well as to guarantee advanced environmental performance. In the case of roadway pavements, the scientific community is focusing on a more fundamental definition of environmental performance by employing the so-called life cycle assessment (LCA) methodology.

Since the first publication on this topic in the mid-1990s, LCA has gained increasing attention because it offers a comprehensive methodology for examining the net environmental performance of different types of pavements, materials and processes [1]. It should be noted that road pavements necessitate the intensive consumption of non-renewable materials (aggregates, bitumen, cement) and industrial products during the construction and maintenance phases [2]; therefore, the LCA approach can be an effective method to defining "greener" alternatives [3].

LCA follows the prescriptions of specific product category rules (PCR). Several software packages and tools are available to support agencies, producers and researchers in the analysis of specific areas able to improve the environmental impacts of a specific product, project or infrastructure. Despite the availability of such supports, it is important to point out that the comparability of the environmental product declarations (EPDs) depends on the quality of the PCRs, which, when available, should be harmonized for the proper characterization of the materials.

While the literature offers many studies concerning the LCA of flexible and rigid pavements, the aim of this paper was to test improvements of the so-called corrected Environmental Asphalt Rating (EARc) for flexible or asphalt pavements, along with a preliminary extension of such an index for rigid pavements. The methodology took its cue from a previous study carried out by Chiola et al. [4], in which the EARc was structured as an objective measure that combined environmental and structural pavement performance to help the Italian contracting authority Autostrade per l'Italia (ASPI) in assessing technical offers and assigning scores in the tender phase.

Starting from this point, the approach was improved according to the normalization and weighting factors proposed by Sala et al. [5] and implemented in the LCA software SimaPro version 9.5.0.0. The approach proposed in this paper combines not only the environmental indicators but also coefficients related to material performance coming from the field of pavement engineering.

Despite the present study being focused on road pavements, it could be applied to more general use contexts by adapting the performance criteria and laws to all the different contexts wherein an environmental and performance-based analysis is needed.

## 2. Pavement Life Cycle Assessment

A transport infrastructure's life cycle could not be studied without considering pavement materials. Road networks are the most representative transport system, although choices in the field of pavements, including poor planning and constructing decisions or inadequate materials, certainly lead to environmental consequences, such as increased pollution and resource consumption.

Due to increasing concerns related to the release of greenhouse gas emissions into the air, and ground and water pollutants from different sources, the pavement community is actively committed to researching and exploring low-impact materials and technologies with reduced energy demand. For both rigid and flexible pavements, a variety of solutions is available, such as the inclusion of recycled materials and bio-binders, the introduction of renewable energy sources in the production and construction stages, and the use of a low production temperature.

In the following chapter, the literature from the field of LCA studies on rigid and flexible pavements is presented with a focus on a comparison between these two solutions.

### 2.1. Comparison between Rigid and Flexible Pavements

Flexible pavements are composed of bituminous mixtures on the top layers and a granular foundation placed on the natural subgrade. Such a structure is the most versatile and is characterized by low production and construction costs. However, it requires frequent maintenance. In the case of heavy loading (highly trafficked motorways and airports) or extreme low-temperature conditions, rigid pavements can be a viable alternative. Rigid pavements consist of a concrete slab on the top (with or without steel reinforcement and joints), and a granular or cement-treated subbase placed on the natural subgrade. While this construction approach reduces maintenance costs and improves structural performance, the construction phases and end-of-life operations can be expensive.

Even if the definition of pavement structures depends on specific project characteristics, a comparison between the environmental performances of both solutions could assist in the choice of the best alternative. In this sense, many studies described in the literature are helpful in identifying the potential environmental benefits and burdens of both alternatives.

Zapata and Gambatese [6] compared the energy consumed during the construction of a continuously reinforced concrete pavement (CRCP) versus an asphalt pavement using an abbreviated life cycle inventory assessment. In CRCP, energy is primarily consumed during the manufacturing of cement and reinforcing steel, which together account for approximately 94% of the total amount of energy consumed from the extraction of raw materials to the placement of the CRCP. For asphalt pavements, the major energy consump-

tion occurs during asphalt mixing and the drying of aggregates (48%), and the production of bitumen (40%).

A similar assessment was performed for the SR 836-Southwest Extension Project in Miami, Florida [7]: when all phases are taken into consideration, rigid pavements have lower environmental impacts and costs, while flexible pavement designs report lower cost and environmental impacts only for the construction phase.

Arpad and Hendrickson [8] compared asphalt pavements and steel-reinforced concrete pavements in terms of their initial construction; they found that the former showed higher energy input and lower ore and fertilizer input requirements and toxic emissions, but also showed high associated hazardous waste generation and management.

In a study focused on the generation of solid waste, Rajendran and Gambatese [9] compared continuously reinforced concrete (CRC) and asphalt concrete (AC) pavements. While the waste created during the placement of materials is almost negligible compared to that generated during the manufacturing and end-of-life (EOL) phases for both pavement structures, waste generation represents around 53% and 59% of the total waste production at the EOL phase for CRC and AC pavements, respectively.

Aurangzed et al. [10] conducted a hybrid LCA, comprising a combination of process-based LCA and economic input–output LCA (EIO-LCA), covering the material production, construction, maintenance and rehabilitation phases of a pavement, thus addressing its entire life-cycle. They suggest that care should be carried out when flexible and rigid pavements are compared in terms of energy consumption, because the inclusion of the feedstock energy scale heavily favors rigid pavements.

For the evaluation of the potential impacts of recycled materials on the construction of urban roads [11,12], LCA studies were carried out, analyzing asphalt mixtures containing different amounts of reclaimed asphalt (RA). The results show that such mixtures increase the consumption of energy but lead to a reduction in the nr-CED and GWP associated with the production of asphalt materials and enhance the environmental performance of road pavements.

In most of the cases, all the abovementioned results clearly suggest that improving the sustainability strategy (reduce, recover, reuse, and recycle waste) and using recycled instead of natural materials would help to improve sustainability in the pavement sector. On the other hand, in the literature, several approaches to the sustainability problem can be found, but this makes it difficult to define a final evaluation criterion. For this reason, the methodology presented in this study uses the limited set of parameters included in EPD certifications, which are also popular for use on pavements in the characterization of road materials and pavement structures (refer to Table 1).

**Table 1.** Impact categories.

| Core Environmental Indicators (A) | Abbreviation | Unit of Meas. |
|---|---|---|
| Climate change | GWP | kg $CO_2$ eq. |
| Ozone depletion | ODP | Kg $CFC_{11}$ eq. |
| Acidification | AP | mol $H^+$ eq. |
| Eutrophication aquatic freshwater | EP, f | kg P eq. |
| Eutrophication aquatic marine | EP, m | kg N eq. |
| Eutrophication terrestrial | EP, t | mol N eq. |
| Photochemical ozone formation | POCP | kg NMVOC eq. |
| Depletion of abiotic resources—minerals and metals | ADPE | kg Sb eq. |
| Depletion of abiotic resources—fossil fuels | ADPF | MJ |
| Water use | WDP | $m^3$ |
| Use of resources (B) | | |
| Use of renewable energy not as raw material | PERE | MJ |
| Use of renewable energy as raw materials | PERM | MJ |
| Total renewable energy | PERT | MJ |

**Table 1.** *Cont.*

| Use of resources (B) | | |
|---|---|---|
| Use of non-renewable energy not as raw materials | PENRE | MJ |
| Use of non-renewable energy as raw materials | PENRM | MJ |
| Total non-renewable energy | PENRT | MJ |
| Use of secondary material | SM | kg |
| Use of renewable secondary fuels | RSF | MJ |
| Use of non-renewable secondary fuels | NRF | MJ |
| Net fresh water | NFW | $m^3$ |
| Waste categories and output flows (C) | | |
| Hazardous waste disposed | HWD | kg |
| Non-hazardous waste disposed | NHWD | kg |
| Radioactive waste disposed | RWD | kg |
| Components for re-use | CRU | kg |
| Components for recycling | MFR | kg |
| Materials for energy recovery | MER | kg |
| Exported energy | EE | MJ |

Moreover, an analysis based only on environmental criteria without considering the pavement's performance only represents a partial evaluation of the environmental benefits and drawbacks. Accordingly, the existing EAR index [4], which already includes considerations of the design life and the mechanical performance for flexible asphalt pavements, was extended and fine-tuned for application to rigid pavements.

Combination of Environmental and Performance Indicators

The EAR index represents the first attempt to combine environmental and structural performance under the same indicator. To the best of the authors' knowledge, no other similar or equivalent model has been found in the literature, although it should be mentioned that other indicators were developed to support road agencies in the choice of correct materials or construction strategies.

The first examples were found in 2013, with the commencement of the EDGAR project [13] founded by the EU with the aim of supporting the National Road Administrations in the sustainability assessment of novel green bituminous mixtures, in terms of both the environmental aspects, which are predominantly addressed by the EPD declaration, but also socio-economic factors and a long-term view of sustainability aspects. In the same year, the LICCER model was presented, building from the Norwegian EFFEKT model [14], with the objective being to provide planners with quantitative information regarding road corridor alternatives related to environmental aspects [15]. Regarding the comparison of different maintenance operations, it is important to mention the HERMES project, which developed a methodology enabling the selection of the best available technology and strategy with the lowest costs for the environment and society. Addressing cementitious-based construction materials, but without specific reference to road pavements as of yet, the EASI coefficient [16] should also be mentioned.

In Section 3, the revised EAR methodology is presented, with a focus on the main revisions and the improvements implemented related to the updated version of the ISO standard. In Section 4, the new preliminarily approach for rigid pavement design is described.

## 3. The Environmental Asphalt Rate (EAR)—Revision of the Method

Considering that road pavements are a very significant infrastructural asset in terms of construction materials and resource consumption, the adoption of a sustainable management approach is becoming a priority in order to reduce the environmental impacts of road construction and maintenance treatments.

Given the need to compare between alternative pavement maintenance techniques, a calculation methodology has been presented by Chiola et al. [4] that, based on the

information that can be extracted from EPD certifications, yields a single index in the form of EAR, which can be used to summarize all the different aspects of environmental impacts for both the production and paving phases of asphalt mixtures.

Moreover, the definition of performance parameters to be applied to the EAR index was carried out in order to obtain an overall index, named EARc, which also includes the structural and functional properties of a flexible pavement. Even though the EARc index has been included in the ASPI Pavement Management System (PMS), it has not yet been used in the tender evaluation phase. Some updates of the EAR calculation method have been made, as described in the rest of this chapter.

### 3.1. Environmental Product Declaration EPD

The EPD certification is described in the standards ISO 14025 [17] and EN 15804 [18]. These standards allow for the quantification of the environmental impacts related to the entire life cycle of a material. The assessment of environmental impacts is carried out through the determination of several impact categories, defined by Environmental Performance Indicators. Building from [4], the list of these indicators has recently been updated. The latest update of the preliminary list, referred to as Version 2.0, is described below, by means of a summary table (Table 1) for each impact category: (A) core environmental indicators, (B) consumption of resources intended for use as raw materials and as energy, and (C) the generation of waste and the management of different output flows. In (B), the environmental indicators PENRE and PENRT include feedstock energy.

In this paper, the evaluation of the EAR coefficient is carried out with reference to this updated list of indicators, as will be thoroughly described in the rest of this chapter. The life cycle phases considered here are those associated with Modules A1–A5.

### 3.2. Data Collection

Publicly available EPD certifications were used to collect data on asphalt mixtures. All these EPDs were produced according to the Product Category Rules (PCR) [19]. These certifications have been used to define the environmental impacts related to phases A1, A2, and A3 (i.e., raw materials supply, transport and manufacturing of asphalt).

The EPD certifications considered in this paper refer to the updated list of Environmental Performance Indicators (Version 2.0) previously described.

Only mixtures produced in Sweden were selected at this stage, mainly because of the greater availability of updated EPDs adopting the new set of indicators, but also to exclude possible differences related only to the geographical origins of the mixtures. The names of the plants and references to the EPD certification are listed in the following works: Porsen [20], Kärra [21], Arlanda [22], Riksten [23], Gräfsåsen [24], Grönsberg [25], Hudiksvall [26], Ramnaslätt [27], Skellefteå [28] and Växbo [29].

The evaluations related to phases A4 and A5 (i.e., the transport and laying of the asphalt concrete at the site) were carried out as described in [4], considering the same reference scenario and making the same assumptions, but adopting the updated list of Environmental Performance Indicators (Version 2.0).

The data employed for assigning the previously mentioned EPD certifications and carrying out the construction process stages' impact estimations are reported in Table 2.

In this section, the EAR index is calculated for each mixture, following the same weighting and normalization approach as described in [4] but applying it to the updated set of Environmental Performance Indicators (Version 2.0).

**Table 2.** Environmental Performance Indicator values (modules A1–A5) per 1 ton of asphalt mixture. Calculation of EAR index with simplified weights and normalization factors.

| Indicator | Unit of Meas. | Porsen | Kärra | Arlanda | Riksten | Gräfsåsen | Grönsberg | Hudiksvall | Ramnaslätt | Skellefteå | Växbo |
|---|---|---|---|---|---|---|---|---|---|---|---|
| | | | | | Core environmental indicators (A) | | | | | | |
| GWP, Tot | kg $CO_2$ eq. | $3.64 \times 10^{+01}$ | $3.64 \times 10^{+01}$ | $2.74 \times 10^{+01}$ | $3.34 \times 10^{+01}$ | $3.04 \times 10^{+01}$ | $3.14 \times 10^{+01}$ | $4.54 \times 10^{+01}$ | $2.34 \times 10^{+01}$ | $2.74 \times 10^{+01}$ | $2.84 \times 10^{+01}$ |
| ODP | kg CFC11 eq. | $2.90 \times 10^{-11}$ | $1.70 \times 10^{-14}$ | $4.50 \times 10^{-08}$ | $8.40 \times 10^{-08}$ | $7.60 \times 10^{-08}$ | $2.50 \times 10^{-11}$ | $8.30 \times 10^{-08}$ | $3.90 \times 10^{-08}$ | $1.30 \times 10^{-07}$ | $8.20 \times 10^{-08}$ |
| AP | mol $H^+$ eq. | $2.27 \times 10^{-01}$ | $2.17 \times 10^{-01}$ | $1.57 \times 10^{-01}$ | $2.37 \times 10^{-01}$ | $2.17 \times 10^{-01}$ | $2.07 \times 10^{-01}$ | $2.07 \times 10^{-01}$ | $1.57 \times 10^{-01}$ | $1.87 \times 10^{-01}$ | $2.27 \times 10^{-01}$ |
| EP, f | kg P eq. | $3.80 \times 10^{-04}$ | $3.70 \times 10^{-04}$ | $4.50 \times 10^{-04}$ | $7.80 \times 10^{-04}$ | $6.30 \times 10^{-04}$ | $4.20 \times 10^{-04}$ | $3.30 \times 10^{-04}$ | $5.20 \times 10^{-04}$ | $5.10 \times 10^{-04}$ | $3.20 \times 10^{-04}$ |
| EP, m | kg N eq. | $7.32 \times 10^{-02}$ | $6.52 \times 10^{-02}$ | $5.52 \times 10^{-02}$ | $7.62 \times 10^{-02}$ | $7.52 \times 10^{-02}$ | $6.62 \times 10^{-02}$ | $6.42 \times 10^{-02}$ | $5.62 \times 10^{-02}$ | $5.92 \times 10^{-02}$ | $6.62 \times 10^{-02}$ |
| EP, t | mol N eq. | $6.49 \times 10^{-01}$ | $5.69 \times 10^{-01}$ | $4.49 \times 10^{-01}$ | $6.39 \times 10^{-01}$ | $6.39 \times 10^{-01}$ | $5.79 \times 10^{-01}$ | $5.59 \times 10^{-01}$ | $4.49 \times 10^{-01}$ | $4.89 \times 10^{-01}$ | $5.99 \times 10^{-01}$ |
| POCP | kg NMVOC eq. | $1.81 \times 10^{-01}$ | $1.61 \times 10^{-01}$ | $1.11 \times 10^{-01}$ | $1.71 \times 10^{-01}$ | $1.71 \times 10^{-01}$ | $1.61 \times 10^{-01}$ | $1.51 \times 10^{-01}$ | $1.11 \times 10^{-01}$ | $1.11 \times 10^{-01}$ | $1.51 \times 10^{-01}$ |
| ADPE | kg Sb eq. | $2.70 \times 10^{-06}$ | $2.70 \times 10^{-06}$ | $1.80 \times 10^{-05}$ | $3.20 \times 10^{-05}$ | $2.80 \times 10^{-05}$ | $2.50 \times 10^{-06}$ | $3.00 \times 10^{-05}$ | $1.60 \times 10^{-05}$ | $4.50 \times 10^{-05}$ | $2.90 \times 10^{-05}$ |
| ADPF | MJ | $2.90 \times 10^{+03}$ | $2.98 \times 10^{+03}$ | $1.64 \times 10^{+03}$ | $2.65 \times 10^{+03}$ | $2.32 \times 10^{+03}$ | $2.61 \times 10^{+03}$ | $2.47 \times 10^{+03}$ | $1.33 \times 10^{+03}$ | $1.78 \times 10^{+03}$ | $2.30 \times 10^{+03}$ |
| WDP | $m^3$ | $6.40 \times 10^{+00}$ | $6.40 \times 10^{+00}$ | $3.90 \times 10^{+00}$ | $6.40 \times 10^{+00}$ | $4.80 \times 10^{+00}$ | $5.60 \times 10^{+00}$ | $2.30 \times 10^{+00}$ | $4.10 \times 10^{+00}$ | $7.10 \times 10^{-01}$ | $9.70 \times 10^{-01}$ |
| | | | | | Use of resources (B) | | | | | | |
| PERE | MJ | $3.94 \times 10^{+02}$ | $3.64 \times 10^{+02}$ | $2.57 \times 10^{+02}$ | $4.58 \times 10^{+02}$ | $2.88 \times 10^{+02}$ | $3.37 \times 10^{+02}$ | $7.10 \times 10^{+01}$ | $3.12 \times 10^{+02}$ | $5.80 \times 10^{+01}$ | $8.60 \times 10^{+01}$ |
| PERM | MJ | $4.80 \times 10^{+01}$ | $6.40 \times 10^{+01}$ | $0.00 \times 10^{+00}$ | $8.00 \times 10^{+01}$ | $0.00 \times 10^{+00}$ | $0.00 \times 10^{+00}$ | $0.00 \times 10^{+00}$ | $0.00 \times 10^{+00}$ | $0.00 \times 10^{+00}$ | $0.00 \times 10^{+00}$ |
| PERT | MJ | $4.42 \times 10^{+02}$ | $4.28 \times 10^{+02}$ | $2.57 \times 10^{+02}$ | $5.38 \times 10^{+02}$ | $2.88 \times 10^{+02}$ | $3.37 \times 10^{+02}$ | $7.10 \times 10^{+01}$ | $3.12 \times 10^{+02}$ | $5.80 \times 10^{+01}$ | $8.60 \times 10^{+01}$ |
| PENRE | MJ | $4.11 \times 10^{+02}$ | $3.95 \times 10^{+02}$ | $3.31 \times 10^{+02}$ | $3.89 \times 10^{+02}$ | $3.92 \times 10^{+02}$ | $3.58 \times 10^{+02}$ | $5.61 \times 10^{+02}$ | $2.83 \times 10^{+02}$ | $3.31 \times 10^{+02}$ | $3.27 \times 10^{+02}$ |
| PENRM | MJ | $2.62 \times 10^{+03}$ | $2.72 \times 10^{+03}$ | $1.44 \times 10^{+03}$ | $2.40 \times 10^{+03}$ | $2.10 \times 10^{+03}$ | $2.39 \times 10^{+03}$ | $2.04 \times 10^{+03}$ | $1.17 \times 10^{+03}$ | $1.58 \times 10^{+03}$ | $2.10 \times 10^{+03}$ |
| PENRT | MJ | $3.03 \times 10^{+03}$ | $3.11 \times 10^{+03}$ | $1.77 \times 10^{+03}$ | $2.78 \times 10^{+03}$ | $2.49 \times 10^{+03}$ | $2.74 \times 10^{+03}$ | $2.60 \times 10^{+03}$ | $1.45 \times 10^{+03}$ | $1.91 \times 10^{+03}$ | $2.43 \times 10^{+03}$ |
| SM | kg | $1.86 \times 10^{+02}$ | $1.43 \times 10^{+02}$ | $3.78 \times 10^{+02}$ | $2.18 \times 10^{+02}$ | $2.32 \times 10^{+02}$ | $2.85 \times 10^{+02}$ | $3.65 \times 10^{+02}$ | $3.58 \times 10^{+02}$ | $9.62 \times 10^{+02}$ | $3.02 \times 10^{+02}$ |
| RSF | MJ | $0.00 \times 10^{+00}$ | $0.00 \times 10^{+00}$ | $0.00 \times 10^{+00}$ | $0.00 \times 10^{+00}$ | $0.00 \times 10^{+00}$ | $0.00 \times 10^{+00}$ | $0.00 \times 10^{+00}$ | $0.00 \times 10^{+00}$ | $0.00 \times 10^{+00}$ | $0.00 \times 10^{+00}$ |
| NRSF | MJ | $0.00 \times 10^{+00}$ | $0.00 \times 10^{+00}$ | $0.00 \times 10^{+00}$ | $0.00 \times 10^{+00}$ | $0.00 \times 10^{+00}$ | $0.00 \times 10^{+00}$ | $0.00 \times 10^{+00}$ | $0.00 \times 10^{+00}$ | $0.00 \times 10^{+00}$ | $0.00 \times 10^{+00}$ |
| FW | $m^3$ | $3.50 \times 10^{-01}$ | $3.20 \times 10^{-01}$ | $2.20 \times 10^{-01}$ | $2.80 \times 10^{-01}$ | $2.00 \times 10^{-01}$ | $2.40 \times 10^{-01}$ | $2.00 \times 10^{-01}$ | $2.10 \times 10^{-01}$ | $1.40 \times 10^{-01}$ | $2.40 \times 10^{-01}$ |
| | | | | | Output flows and waste (C) | | | | | | |
| HWD | kg | $8.80 \times 10^{-02}$ | $7.20 \times 10^{-02}$ | $1.40 \times 10^{-02}$ | $9.20 \times 10^{-03}$ | $8.50 \times 10^{-03}$ | $4.60 \times 10^{-03}$ | $1.10 \times 10^{-02}$ | $2.10 \times 10^{-03}$ | $1.10 \times 10^{-02}$ | $6.40 \times 10^{-03}$ |
| NHWD | kg | $5.20 \times 10^{-01}$ | $6.20 \times 10^{-01}$ | $3.00 \times 10^{-01}$ | $6.80 \times 10^{-01}$ | $3.30 \times 10^{-01}$ | $3.70 \times 10^{-01}$ | $9.60 \times 10^{-02}$ | $6.50 \times 10^{-01}$ | $2.00 \times 10^{-01}$ | $5.80 \times 10^{-02}$ |
| RWD | kg | $8.80 \times 10^{-04}$ | $1.20 \times 10^{-03}$ | $5.40 \times 10^{-04}$ | $9.90 \times 10^{-04}$ | $5.40 \times 10^{-04}$ | $6.40 \times 10^{-04}$ | $7.50 \times 10^{-04}$ | $5.50 \times 10^{-04}$ | $1.80 \times 10^{-04}$ | $2.50 \times 10^{-04}$ |
| CRU | kg | $0.00 \times 10^{+00}$ | $0.00 \times 10^{+00}$ | $0.00 \times 10^{+00}$ | $0.00 \times 10^{+00}$ | $0.00 \times 10^{+00}$ | $0.00 \times 10^{+00}$ | $0.00 \times 10^{+00}$ | $0.00 \times 10^{+00}$ | $0.00 \times 10^{+00}$ | $0.00 \times 10^{+00}$ |
| MFR | kg | $1.10 \times 10^{-01}$ | $1.20 \times 10^{-01}$ | $3.70 \times 10^{-02}$ | $1.90 \times 10^{-01}$ | $4.80 \times 10^{-02}$ | $3.90 \times 10^{-02}$ | $0.00 \times 10^{+00}$ | $2.50 \times 10^{-01}$ | $1.30 \times 10^{-01}$ | $2.50 \times 10^{-03}$ |
| MER | kg | $1.10 \times 10^{-01}$ | $1.80 \times 10^{-01}$ | $5.10 \times 10^{-02}$ | $1.70 \times 10^{-01}$ | $3.20 \times 10^{-02}$ | $1.60 \times 10^{-02}$ | $2.50 \times 10^{-02}$ | $8.00 \times 10^{-02}$ | $5.40 \times 10^{-02}$ | $2.50 \times 10^{-02}$ |
| EE | MJ | $0.00 \times 10^{+00}$ | $0.00 \times 10^{+00}$ | $0.00 \times 10^{+00}$ | $0.00 \times 10^{+00}$ | $0.00 \times 10^{+00}$ | $0.00 \times 10^{+00}$ | $0.00 \times 10^{+00}$ | $0.00 \times 10^{+00}$ | $0.00 \times 10^{+00}$ | $0.00 \times 10^{+00}$ |

The values of the weighting factors have been identified via the assessment of the most significant impacts of the entire asphalt production process. The water use indicator (WDP) has been assigned a weight of zero for the same reason that the freshwater use indicator has been, i.e., because the current technology used for the measurement of water consumption is highly error-prone. Nevertheless, the index has been defined so that this parameter can also be included in the future following the required advances in measurement technologies. For all indicators, the value of the normalization factor has been assumed to be equal to the maximum indicator value yielded by the investigated samples of EPD-certified asphalt mixtures.

Table 3 reports the weighting ($w_f$) and normalization factors ($V_n$).

**Table 3.** Simplified values for weighting and normalization factors.

| Indicator | Weighting Factor $w_f$ | Normalization Factor $V_n$ | Normalization Factor Unit of Measurement |
|---|---|---|---|
| Core environmental indicators (A) | | | |
| GWP, Total | 3.00 | $3.64 \times 10^{+01}$ | kg $CO_2$ eq. |
| ODP | 1.00 | $8.40 \times 10^{-08}$ | kg CFC11 eq. |
| AP | 1.00 | $2.37 \times 10^{-01}$ | mol $H^+$ eq. |
| EP, freshwater | 0.33 | $7.80 \times 10^{-04}$ | kg P eq. |
| EP, marine | 0.33 | $7.62 \times 10^{-02}$ | kg N eq. |
| EP, terrestrial | 0.33 | $6.49 \times 10^{-01}$ | mol N eq. |
| POCP | 1.00 | $1.81 \times 10^{-01}$ | kg NMVOC eq. |
| ADPE | 1.00 | $3.20 \times 10^{-05}$ | kg Sb eq. |
| ADPF | 2.00 | $2.98 \times 10^{+03}$ | MJ |
| WDP | 0.00 | $6.40 \times 10^{+00}$ | $m^3$ |
| Use of resources (B) | | | |
| PERE | 0.00 | $4.58 \times 10^{+02}$ | MJ |
| PERM | 1.43 | $8.00 \times 10^{+01}$ | MJ |
| PERT | 1.43 | $5.38 \times 10^{+02}$ | MJ |
| PENRE | 0.00 | $4.11 \times 10^{+02}$ | MJ |
| PENRM | 1.43 | $2.72 \times 10^{+03}$ | MJ |
| PENRT | 1.43 | $3.11 \times 10^{+03}$ | MJ |
| SM | 1.43 | $3.78 \times 10^{+02}$ | kg |
| RSF | 1.43 | $0.00 \times 10^{+00}$ | MJ |
| NRSF | 1.43 | $0.00 \times 10^{+00}$ | MJ |
| FW | 0.00 | $3.50 \times 10^{-01}$ | $m^3$ |
| Output flows and waste (C) | | | |
| HWD | 1.00 | $8.80 \times 10^{-02}$ | kg |
| NHWD | 1.00 | $6.80 \times 10^{-01}$ | kg |
| RWD | 1.00 | $1.20 \times 10^{-03}$ | kg |
| CRU | 0.00 | $0.00 \times 10^{+00}$ | kg |
| MFR | 2.50 | $1.90 \times 10^{-01}$ | kg |
| MER | 2.50 | $1.80 \times 10^{-01}$ | kg |
| EE | 2.00 | $0.00 \times 10^{+00}$ | MJ |

Following the same procedure as that illustrated in [4], EAR index values were calculated for all the EPD-certified mixtures analyzed in this study, resulting in the values shown Figure 1. The EAR values obtained vary from 47.92 to 81.32, demonstrating good variability in the results.

### 3.3. Calculation of EAR Index with SimaPro Weights and Normalization Factors

In this section, the EAR index is calculated for each asphalt mixture following a different weighting and normalization approach.

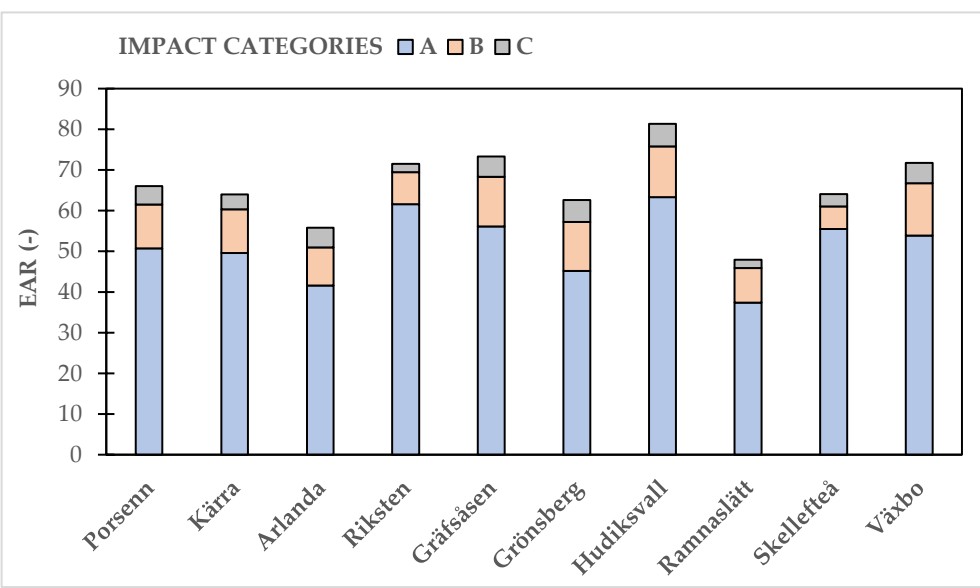

**Figure 1.** EAR values calculated with the simplified weighting and normalization factors for impact categories.

The values of the weighting and normalization factors are extracted via an impact assessment method based on that shown in [5] and performed in the SimaPro software version 9.5.0.0.

In this case, only the core environmental indicators are considered, while all the other indicators are assigned a null weighting factor. Both the weighting and normalization factors are reported in Table 4.

**Table 4.** Simplified values of weighting and normalization factors.

| Indicator | Weighting Factor $w_f$ | Normalization Factor $V_n$ | Normalization Factor Unit of Measurement |
|---|---|---|---|
| Core environmental indicators (A) | | | |
| GWP, Total | $2.11 \times 10^{-01}$ | $8.10 \times 10^{+03}$ | kg $CO_2$ eq. |
| ODP | $6.31 \times 10^{-02}$ | $5.36 \times 10^{-02}$ | kg CFC11 eq. |
| AP | $6.20 \times 10^{-02}$ | $5.56 \times 10^{+01}$ | mol $H^+$ eq. |
| EP, freshwater | $2.80 \times 10^{-02}$ | $1.61 \times 10^{+00}$ | kg P eq. |
| EP, marine | $2.96 \times 10^{-02}$ | $1.95 \times 10^{+01}$ | kg N eq. |
| EP, terrestrial | $3.71 \times 10^{-02}$ | $1.77 \times 10^{+02}$ | mol N eq. |
| POCP | $4.78 \times 10^{-02}$ | $4.06 \times 10^{+01}$ | kg NMVOC eq. |
| ADPE | $7.55 \times 10^{-02}$ | $6.37 \times 10^{-02}$ | kg Sb eq. |
| ADPF | $8.32 \times 10^{-02}$ | $6.50 \times 10^{+04}$ | MJ |
| WDP | $8.51 \times 10^{-02}$ | $1.15 \times 10^{+04}$ | $m^3$ |
| Use of resources (B) | | | |
| PERE | $0.00 \times 10^{+00}$ | $0.00 \times 10^{+00}$ | MJ |
| PERM | $0.00 \times 10^{+00}$ | $0.00 \times 10^{+00}$ | MJ |
| PERT | $0.00 \times 10^{+00}$ | $0.00 \times 10^{+00}$ | MJ |
| PENRE | $0.00 \times 10^{+00}$ | $0.00 \times 10^{+00}$ | MJ |
| PENRM | $0.00 \times 10^{+00}$ | $0.00 \times 10^{+00}$ | MJ |
| PENRT | $0.00 \times 10^{+00}$ | $0.00 \times 10^{+00}$ | MJ |
| SM | $0.00 \times 10^{+00}$ | $0.00 \times 10^{+00}$ | kg |
| RSF | $0.00 \times 10^{+00}$ | $0.00 \times 10^{+00}$ | MJ |
| NRSF | $0.00 \times 10^{+00}$ | $0.00 \times 10^{+00}$ | MJ |
| FW | $0.00 \times 10^{+00}$ | $0.00 \times 10^{+00}$ | $m^3$ |

**Table 4.** *Cont.*

| Indicator | Weighting Factor $w_f$ | Normalization Factor $V_n$ | Normalization Factor Unit of Measurement |
|:---:|:---:|:---:|:---:|
| | Output flows and waste (C) | | |
| HWD | $0.00 \times 10^{+00}$ | $0.00 \times 10^{+00}$ | kg |
| NHWD | $0.00 \times 10^{+00}$ | $0.00 \times 10^{+00}$ | kg |
| RWD | $0.00 \times 10^{+00}$ | $0.00 \times 10^{+00}$ | kg |
| CRU | $0.00 \times 10^{+00}$ | $0.00 \times 10^{+00}$ | kg |
| MFR | $0.00 \times 10^{+00}$ | $0.00 \times 10^{+00}$ | kg |
| MER | $0.00 \times 10^{+00}$ | $0.00 \times 10^{+00}$ | kg |
| EE | $0.00 \times 10^{+00}$ | $0.00 \times 10^{+00}$ | MJ |

Following the same procedure as was illustrated in [4], EAR index values have been calculated for all EPD-certified mixtures analyzed in this study, resulting in the values shown in Figure 2. The EAR values obtained vary from 0.0185 to 0.0355, demonstrating good variability in the results.

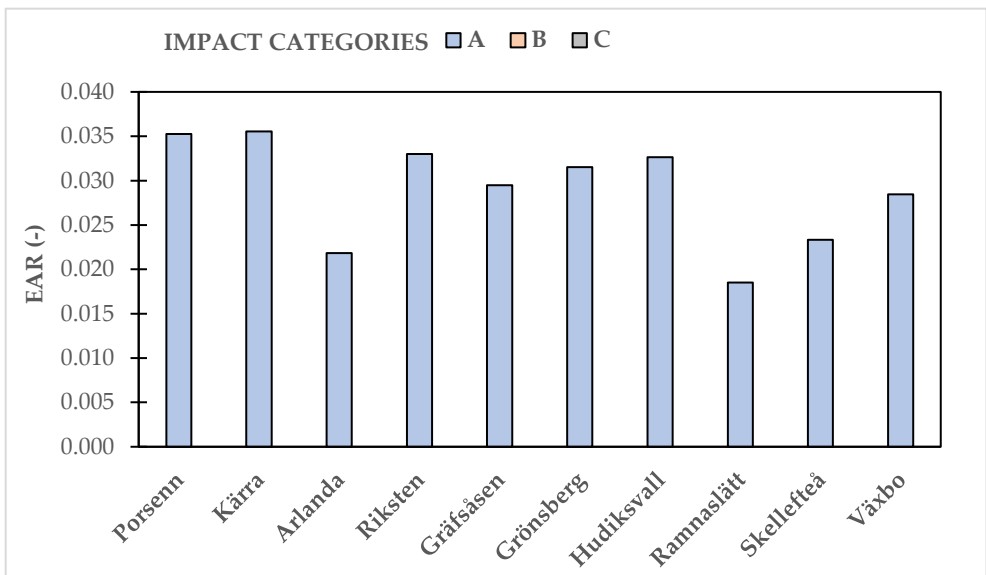

**Figure 2.** EAR values calculated with the SimaPro weighting and normalization factors (only for impact category A).

*3.4. Comparisons between Results derived with Simplified and SimaPro Weights and Normalization Factors*

It is possible to compare the results obtained for the EAR index via either the simplified or the SimaPro weighting and normalization factors.

One clear difference concerns the order of magnitude of the EAR values obtained. This directly depends on the different definition criteria employed for the weighting and normalization factors.

However, it is possible to observe a certain similarity between the two methods in terms of the trend of the EAR index among the different mixtures.

To efficiently compare the two methods, it is possible to apply a second normalization to the EAR values, dividing the EAR value of each mixture by the maximum EAR value obtained from the entire set of mixtures. This is conducted separately for the set of EAR values obtained with the ASPI factors and for the set of EAR values obtained with the SimaPro factors.

The results of this elaboration are reported in Figure 3. In this figure, in order to yield a comparison between the two approaches with respect to the normalization

method applied, the data refer only to impact category A, where both approaches include normalization factors.

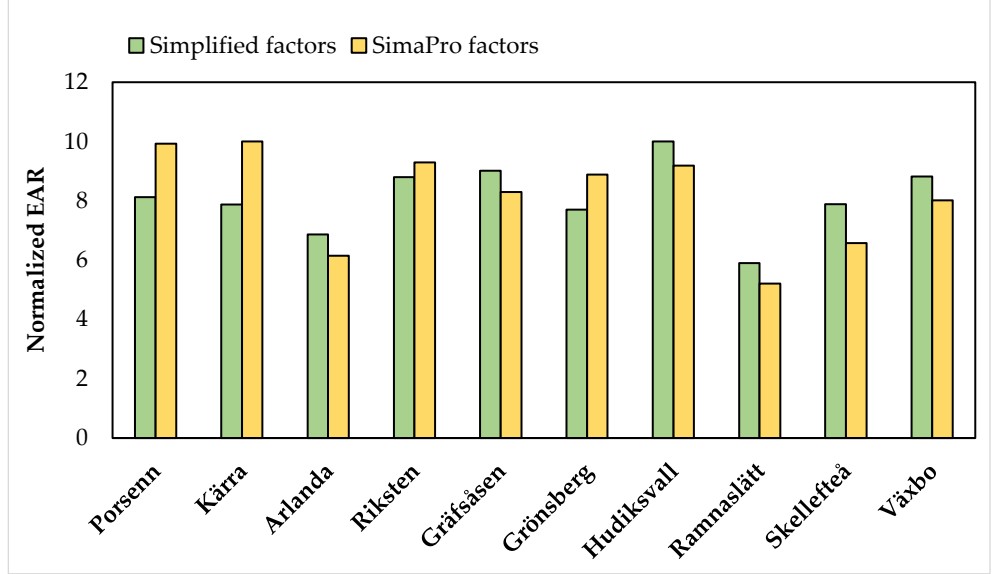

**Figure 3.** Comparison of normalized EAR values calculated with simplified and SimaPro weighting and normalization factors.

The graph highlights the similarities in the results obtained with the two weighting and normalization methods. These can be summarized as follows:

- The Ramnaslätt mixture performs better than the others, yielding the lowest value of the EAR index;
- Very similar environmental impact is estimated for the Porsenn and Kärra mixtures.

At the same time, it is possible to observe the following difference; according to the simplified method, the Hudiksvall mixture is the one with the lowest performance, since it is the one showing the highest EAR value. On the contrary, the SimaPro factors lead to an EAR value for the Hudiksvall mixture that is lower than the values obtained for the Porsenn and Kärra mixtures, the ones reporting the highest environmental im-pact in this case.

Overall, the results obtained with the two sets of weighting and normalization factors are consistent with each other. Having said that, it is important to note that the SimaPro weighting and normalization factors are based on a much larger library than the factors adopted via the simplified method, which were calculated via a basic procedure, and that they can be extracted by anyone who has access to the SimaPro dataset. On one hand, this suggests that the simplified method is straightforward but is still suitable for determining the EAR index. On the other hand, it is possible to conclude that the factors provided by SimaPro, or other specific LCA software, can be exploited to obtain a well-founded and shareable estimation of the EAR index.

### 3.5. Definition of the Performance Coefficients

As already mentioned at the beginning of this section, a series of corrective performance coefficients that can be applied to the EAR index were defined in [4], and these allow for the calculation of EARc while also taking into account the mechanical performance and the service life of asphalt mixtures.

The definition of these coefficients is based on the pre-qualification data of the mixtures, obtained from the verification and acceptance procedure for bituminous mixtures, employing models that allow one to quantify the most significant performance aspects for each type of mixture. In particular, the following three performance coefficient were defined:

- TDC Coefficient (I1) for open-graded surface mixtures, addressing the behavior with respect to the top-down cracking phenomenon;
- Fatigue Coefficient (I2) for dense graded mixtures, related to fatigue behavior;
- Noise Coefficient (I3) for both open-graded and dense-graded mixtures, related to the acoustic emissions.

The procedure for the calculation of the EARc index consists of applying the different performance indexes I1, I2 and I3 as multiplicative coefficients of the EAR index.

The recent update of the EPDs to Version 2.0, as described previously, which necessitated the update of the calculation procedure of the EAR index here presented, does not alter the definitions of the performance coefficients or the EARc index.

Based on the available data for the EPD-certified mixtures considered in this paper, a direct calculation of the EARc is not possible, since no pre-qualification data are available for the mixtures. However, we should still consider Ref. [4], as it shows a comprehensive example of the numerical application of the proposed methodology for the calculation of the EARc index.

## 4. The Environmental Rate for Concrete Pavement (ECR)

Following the same approach as was used for asphalt pavements, the environmental analysis is here extended to cementitious concrete pavements.

Publicly available EPD [30] certifications were used to collect data on concrete mixtures. These certifications have been used to define the environmental impact due to phases A1, A2, and A3.

The EPD certifications considered in this paper employ the updated list of Environmental Performance Indicators (Version 2.0) described previously. The analysis of stages A4 and A5, which are included in the EAR and refer to the strong impact associated with the maintenance phases of asphalt, is omitted here for the rigid pavement. This is because, for that type, the impact of such phases is negligible if compared with the construction phase.

Only mixtures produced in Italy were selected at this stage, to exclude possible differences related only to the geographical origins of mixtures. The locations of the plants are listed below, and the different concretes from each are distinguished by numbers for reasons of confidentiality:

- Beinasco 1;
- Beinasco 2;
- Civitavecchia 1;
- Civitavecchia 2;
- Civitavecchia 3;
- Falconara 1;
- Falconara 2;
- Falconara 3;
- Falconara 4;
- Falconara 5.

In this case, phases A4 and A5 (i.e., transport and laying of the cementitious concrete at the site) were not taken into consideration.

The data extracted for use by the mentioned EPD certifications and the estimations of construction process stages' impacts are reported in Table 5.

The ECR index (Environmental Concrete Rate) is calculated for each of the selected EPD-certified mixtures following the SimaPro weighting and normalization approach. Both the weighting factors and the normalization factors extracted from SimaPro software version 9.5.0.0 are reported in Table 4. The results regarding the calculation of the ECR index are shown in Figure 4.

**Table 5.** Environmental Performance Indicator values (modules A1–A3) per 1 ton of concrete mixture.

| | | Core environmental indicators (A) | | | | | | | | | |
|---|---|---|---|---|---|---|---|---|---|---|---|
| Indicator | Unit of Meas. | Beinasco 1 | Beinasco 2 | Civitavecchia 1 | Civitavecchia 2 | Civitavecchia 3 | Falconara 1 | Falconara 2 | Falconara 3 | Falconara 4 | Falconara 5 |
| GWP, Total | kg $CO_2$ eq. | $2.65 \times 10^{+02}$ | $3.12 \times 10^{+02}$ | $2.57 \times 10^{+02}$ | $3.43 \times 10^{+02}$ | $3.45 \times 10^{+02}$ | $1.45 \times 10^{+02}$ | $1.74 \times 10^{+02}$ | $2.35 \times 10^{+02}$ | $2.83 \times 10^{+02}$ | $2.83 \times 10^{+02}$ |
| ODP | kg CFC11 eq. | $1.20 \times 10^{-05}$ | $1.24 \times 10^{-05}$ | $1.12 \times 10^{-05}$ | $1.42 \times 10^{-05}$ | $1.43 \times 10^{-05}$ | $8.60 \times 10^{-06}$ | $9.30 \times 10^{-06}$ | $1.10 \times 10^{-05}$ | $1.20 \times 10^{-05}$ | $1.25 \times 10^{-05}$ |
| AP | mol $H^+$ eq. | $6.33 \times 10^{-01}$ | $6.98 \times 10^{-01}$ | $6.93 \times 10^{-01}$ | $9.04 \times 10^{-01}$ | $9.15 \times 10^{-01}$ | $5.52 \times 10^{-01}$ | $6.38 \times 10^{-01}$ | $8.16 \times 10^{-01}$ | $9.45 \times 10^{-01}$ | $9.67 \times 10^{-01}$ |
| EP, f | kg P eq. | $1.36 \times 10^{-02}$ | $1.36 \times 10^{-02}$ | $1.62 \times 10^{-02}$ | $2.21 \times 10^{-02}$ | $2.23 \times 10^{-02}$ | $1.44 \times 10^{-02}$ | $1.77 \times 10^{-02}$ | $2.34 \times 10^{-02}$ | $2.87 \times 10^{-02}$ | $2.80 \times 10^{-02}$ |
| EP, m | kg N eq. | $1.03 \times 10^{-03}$ | $1.02 \times 10^{-03}$ | $1.25 \times 10^{-03}$ | $1.78 \times 10^{-03}$ | $1.80 \times 10^{-03}$ | $1.00 \times 10^{-03}$ | $1.23 \times 10^{-03}$ | $1.61 \times 10^{-03}$ | $1.96 \times 10^{-03}$ | $1.91 \times 10^{-03}$ |
| EP, t | mol N eq. | $1.99 \times 10^{+00}$ | $2.29 \times 10^{+00}$ | $2.22 \times 10^{+00}$ | $2.84 \times 10^{+00}$ | $2.87 \times 10^{+00}$ | $1.61 \times 10^{+00}$ | $1.80 \times 10^{+00}$ | $2.25 \times 10^{+00}$ | $2.53 \times 10^{+00}$ | $2.61 \times 10^{+00}$ |
| POCP | kg NMVOC eq. | $5.19 \times 10^{-01}$ | $5.85 \times 10^{-01}$ | $5.30 \times 10^{-01}$ | $7.01 \times 10^{-01}$ | $7.03 \times 10^{-01}$ | $4.14 \times 10^{-01}$ | $4.54 \times 10^{-01}$ | $5.55 \times 10^{-01}$ | $6.18 \times 10^{-01}$ | $6.31 \times 10^{-01}$ |
| ADPE | kg Sb eq. | $1.50 \times 10^{-04}$ | $1.42 \times 10^{-04}$ | $1.41 \times 10^{-04}$ | $2.20 \times 10^{-04}$ | $2.20 \times 10^{-04}$ | $1.12 \times 10^{-04}$ | $1.40 \times 10^{-04}$ | $1.71 \times 10^{-04}$ | $2.15 \times 10^{-04}$ | $1.92 \times 10^{-04}$ |
| ADPF | MJ | $1.13 \times 10^{+03}$ | $1.17 \times 10^{+03}$ | $1.01 \times 10^{+03}$ | $1.25 \times 10^{+03}$ | $1.27 \times 10^{+03}$ | $8.43 \times 10^{+02}$ | $9.40 \times 10^{+02}$ | $1.15 \times 10^{+03}$ | $1.30 \times 10^{+03}$ | $1.32 \times 10^{+03}$ |
| WDP | $m^3$ | $1.39 \times 10^{+02}$ | $1.35 \times 10^{+02}$ | $1.30 \times 10^{+02}$ | $1.32 \times 10^{+02}$ | $1.32 \times 10^{+02}$ | $1.01 \times 10^{+02}$ | $1.07 \times 10^{+02}$ | $1.15 \times 10^{+02}$ | $1.25 \times 10^{+02}$ | $1.41 \times 10^{+02}$ |
| | | Use of resources (B) | | | | | | | | | |
| Indicator | Unit of Meas. | Beinasco 1 | Beinasco 2 | Civitavecchia 1 | Civitavecchia 2 | Civitavecchia 3 | Falconara 1 | Falconara 2 | Falconara 3 | Falconara 4 | Falconara 5 |
| PERE | MJ | $5.25 \times 10^{+01}$ | $5.28 \times 10^{+01}$ | $1.06 \times 10^{+02}$ | $1.44 \times 10^{+02}$ | $1.44 \times 10^{+02}$ | $3.56 \times 10^{+01}$ | $4.25 \times 10^{+02}$ | $5.44 \times 10^{+00}$ | $6.55 \times 10^{+00}$ | $6.29 \times 10^{+00}$ |
| PERM | MJ | $0.00 \times 10^{+00}$ | $0.00 \times 10^{+00}$ | $0.00 \times 10^{+00}$ | $0.00 \times 10^{+00}$ | $0.00 \times 10^{+00}$ | $0.00 \times 10^{+00}$ | $0.00 \times 10^{+00}$ | $0.00 \times 10^{+00}$ | $0.00 \times 10^{+00}$ | $0.00 \times 10^{+00}$ |
| PERT | MJ | $5.25 \times 10^{+01}$ | $5.28 \times 10^{+01}$ | $1.06 \times 10^{+02}$ | $1.44 \times 10^{+02}$ | $1.44 \times 10^{+02}$ | $3.56 \times 10^{+01}$ | $4.25 \times 10^{+02}$ | $5.44 \times 10^{+00}$ | $6.55 \times 10^{+00}$ | $6.29 \times 10^{+00}$ |
| PENRE | MJ | $1.29 \times 10^{+03}$ | $1.33 \times 10^{+03}$ | $1.25 \times 10^{+03}$ | $1.57 \times 10^{+03}$ | $1.59 \times 10^{+03}$ | $9.56 \times 10^{+02}$ | $1.07 \times 10^{+03}$ | $1.31 \times 10^{+03}$ | $1.48 \times 10^{+03}$ | $1.50 \times 10^{+03}$ |
| PENRM | MJ | $0.00 \times 10^{+00}$ | $0.00 \times 10^{+00}$ | $0.00 \times 10^{+00}$ | $0.00 \times 10^{+00}$ | $0.00 \times 10^{+00}$ | $0.00 \times 10^{+00}$ | $0.00 \times 10^{+00}$ | $0.00 \times 10^{+00}$ | $0.00 \times 10^{+00}$ | $0.00 \times 10^{+00}$ |
| PENRT | MJ | $1.29 \times 10^{+03}$ | $1.33 \times 10^{+03}$ | $1.25 \times 10^{+03}$ | $1.57 \times 10^{+03}$ | $1.59 \times 10^{+03}$ | $9.56 \times 10^{+02}$ | $1.07 \times 10^{+03}$ | $1.31 \times 10^{+03}$ | $1.48 \times 10^{+03}$ | $1.50 \times 10^{+03}$ |
| SM | kg | $7.98 \times 10^{+01}$ | $1.69 \times 10^{+01}$ | $1.31 \times 10^{+02}$ | $1.37 \times 10^{+02}$ | $1.37 \times 10^{+02}$ | $8.64 \times 10^{+02}$ | $6.60 \times 10^{+02}$ | $5.36 \times 10^{+02}$ | $2.06 \times 10^{+02}$ | $2.06 \times 10^{+02}$ |
| RSF | MJ | $1.76 \times 10^{+01}$ | $2.18 \times 10^{+01}$ | $0.00 \times 10^{+00}$ | $0.00 \times 10^{+00}$ | $0.00 \times 10^{+00}$ | $5.26 \times 10^{+01}$ | $6.57 \times 10^{+01}$ | $9.37 \times 10^{+01}$ | $1.15 \times 10^{+02}$ | $1.15 \times 10^{+02}$ |
| NRSF | MJ | $4.78 \times 10^{+01}$ | $5.93 \times 10^{+01}$ | $0.00 \times 10^{+00}$ | $0.00 \times 10^{+00}$ | $0.00 \times 10^{+00}$ | $8.24 \times 10^{+01}$ | $1.03 \times 10^{+02}$ | $1.47 \times 10^{+02}$ | $1.80 \times 10^{+02}$ | $1.80 \times 10^{+02}$ |
| FW | $m^3$ | $3.34 \times 10^{+00}$ | $3.25 \times 10^{+00}$ | $3.17 \times 10^{+00}$ | $3.24 \times 10^{+00}$ | $3.24 \times 10^{+00}$ | $2.61 \times 10^{+00}$ | $2.75 \times 10^{+00}$ | $2.96 \times 10^{+00}$ | $3.20 \times 10^{+00}$ | $3.57 \times 10^{+00}$ |

**Table 5.** *Cont.*

| Indicator | Unit of Meas. | Beinasco 1 | Beinasco 2 | Civitavecchia 1 | Civitavecchia 2 | Civitavecchia 3 | Falconara 1 | Falconara 2 | Falconara 3 | Falconara 4 | Falconara 5 |
|---|---|---|---|---|---|---|---|---|---|---|---|
| | | | | | Output flows and waste (C) | | | | | | |
| HWD | kg | $1.96 \times 10^{-02}$ | $3.68 \times 10^{-01}$ | $4.21 \times 10^{-01}$ | $5.70 \times 10^{-01}$ | $5.70 \times 10^{-01}$ | $8.39 \times 10^{-02}$ | $1.05 \times 10^{-01}$ | $1.49 \times 10^{-01}$ | $1.83 \times 10^{-01}$ | $1.83 \times 10^{-01}$ |
| NHWD | kg | $2.96 \times 10^{-01}$ | $2.44 \times 10^{-02}$ | $2.47 \times 10^{-02}$ | $3.72 \times 10^{-02}$ | $3.72 \times 10^{-02}$ | $5.99 \times 10^{-03}$ | $7.49 \times 10^{-03}$ | $1.07 \times 10^{-03}$ | $1.31 \times 10^{-03}$ | $1.31 \times 10^{-03}$ |
| RWD | kg | $0.00 \times 10^{+00}$ | $0.00 \times 10^{+00}$ | $0.00 \times 10^{+00}$ | $0.00 \times 10^{+00}$ | $0.00 \times 10^{+00}$ | $0.00 \times 10^{+00}$ | $0.00 \times 10^{+00}$ | $0.00 \times 10^{+00}$ | $0.00 \times 10^{+00}$ | $0.00 \times 10^{+00}$ |
| CRU | kg | $8.41 \times 10^{+01}$ | $8.41 \times 10^{+01}$ | $5.04 \times 10^{+01}$ | $5.04 \times 10^{+01}$ | $5.04 \times 10^{+01}$ | $9.89 \times 10^{+01}$ | $9.89 \times 10^{+01}$ | $9.89 \times 10^{+01}$ | $9.89 \times 10^{+01}$ | $9.89 \times 10^{+01}$ |
| MFR | kg | $0.00 \times 10^{+00}$ | $0.00 \times 10^{+00}$ | $0.00 \times 10^{+00}$ | $0.00 \times 10^{+00}$ | $0.00 \times 10^{+00}$ | $0.00 \times 10^{+00}$ | $0.00 \times 10^{+00}$ | $0.00 \times 10^{+00}$ | $0.00 \times 10^{+00}$ | $0.00 \times 10^{+00}$ |
| MER | kg | $0.00 \times 10^{+00}$ | $0.00 \times 10^{+00}$ | $0.00 \times 10^{+00}$ | $0.00 \times 10^{+00}$ | $0.00 \times 10^{+00}$ | $0.00 \times 10^{+00}$ | $0.00 \times 10^{+00}$ | $0.00 \times 10^{+00}$ | $0.00 \times 10^{+00}$ | $0.00 \times 10^{+00}$ |
| EE | MJ | $0.00 \times 10^{+00}$ | $0.00 \times 10^{+00}$ | $0.00 \times 10^{+00}$ | $0.00 \times 10^{+00}$ | $0.00 \times 10^{+00}$ | $0.00 \times 10^{+00}$ | $0.00 \times 10^{+00}$ | $0.00 \times 10^{+00}$ | $0.00 \times 10^{+00}$ | $0.00 \times 10^{+00}$ |

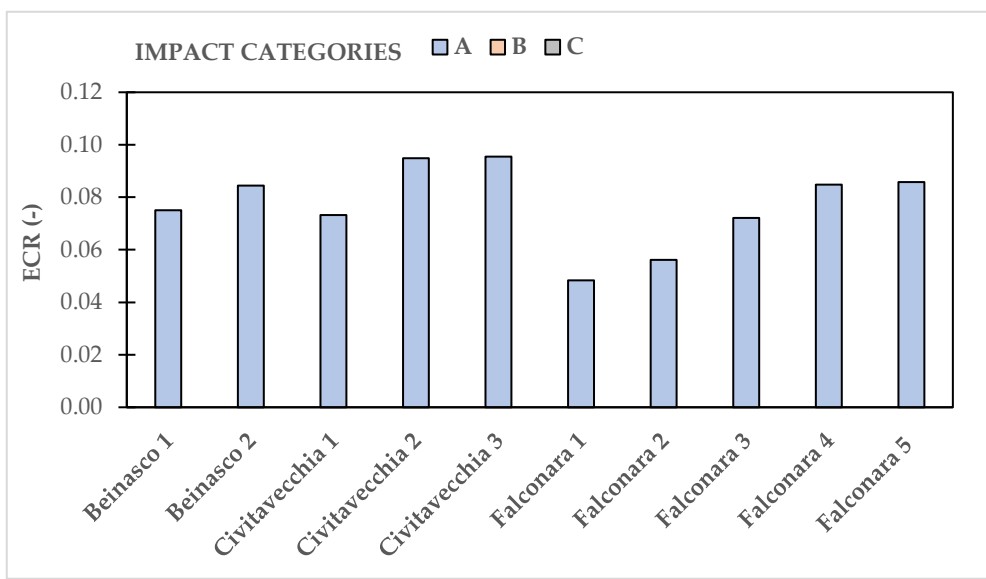

**Figure 4.** ECR values calculated with the SimaPro weighting and normalization factors. Only impact category A is considered.

The obtained ECR values demonstrate a good variability in the results, varying from 0.0483 for the Falconara 1 concrete to 0.0955 for the Civitavecchia 3 concrete. This confirms that the developed approach is generally suitable for assessing the environmental performances of both asphalt and cementitious concrete mixtures.

*4.1. Performance Criteria*

Similar to the $EAR_c$, corrective coefficients ($I_1$, $I_2$, . . ., $I_n$) that can take into account the performance of the cementitious concrete mixtures have been here defined and associated to the results of the environmental analysis. The subsequent corrected Environmental Concrete Rate ($ECR_c$) is defined by Equation (1):

$$ECR_c = ECR \cdot I_1 \cdot I_2 \ldots \cdot I_n \tag{1}$$

It should be considered that corrections ($I_1$, $I_2$, . . ., $I_n$) can be made via a multitude of factors to account for the different performances of the rigid pavement. In general, a pavement structure fulfils structural and functional tasks, the former associated with mechanical resistance and the latter linked to the interaction with traffic. In any case, the definition of such indicators requires an analysis of the rigid pavement behavior when exposed to traffic loads, according to the main theories and methodologies available in the literature.

In the present study, two corrective coefficients are defined that consider the structural ($I_1$) and functional ($I_2$) performance of the concrete slab. While $I_1$ is determined as a function of the fatigue resistance, $I_2$ is based on the roughness of the slab with a focus on the IRI parameter, following the formulation in Equations (2) and (3):

$$I_1 = 1 - \frac{n}{N_f} \tag{2}$$

$$I_2 = \frac{IRI}{IRI_{eol}} \tag{3}$$

where n is the expected number of load repetitions, $N_f$ is the allowable number of load repetitions, IRI is the International Roughness Index, and $IRI_{eol}$ is the maximum value of IRI that can be expected at the end of service life (assumed to be equal to 4.5 mm/m [31]).

The significance of each parameter is highlighted in the following. It should be remembered that rigid pavements are designed by following a mechanist design procedure

based on the definition of a structural model to evaluate the pavement's response and then determine the pavement's state of distress by applying calibrated distress models. After the parameters of the pavement configuration are set (type of pavement, slab geometry, joints, distribution of steel bars, pavement layers, etc.), the data that are input for the structural modeling also include the material properties, the climate model and the traffic composition. The subsequent analysis of pavement response involves the evaluation of the stress, strain, and displacement, which are needed as input for the distress model to predict the resistance against fatigue cracking, pumping, faulting and joint deterioration. Other calculations are performed to analyze thermal and moisture effects (curling and friction at the interface, moisture warping, etc.), as well as stresses and displacements, at some specific points of the slab (corner, edge and center) via Westergaard's formula.

### 4.2. Allowable Number of Load Repetitions

We here focus on a Jointed Reinforced Concrete Pavement (JCRP), a model that could feasibly be applied for Italian motorways. Such a pavement is composed of a jointed reinforced concrete slab, a cemented treated foundation layer (typical elastic modulus of 3000 MPa), and a subgrade that has been carefully prepared before the pavement's construction (typical elastic modulus of 120 MPa). When experimental data are not available, concrete properties can be derived from the 28-day cubic compressive strength ($R_{ck}$) by following the Italian construction standard [32] using Equations (4)–(8) for cylindrical compressive strength ($f_{ck}$), cubic mean compressive strength ($f_{cm}$), flexural axial strength ($f_{ctm}$), flexural tensile strength ($f_{cfm}$) and the elastic modulus ($E_{cm}$):

$$f_{ck} = 0.83\,R_{ck} \tag{4}$$

$$f_{cm} = f_{ck} + 8 \tag{5}$$

$$f_{cm} = f_{ck} + 8 \tag{6}$$

$$f_{cfm} = 1.2 f_{ctm} \tag{7}$$

$$E_{cm} = 22000 \left( f_{cm}/10 \right)^{0.3} \tag{8}$$

As an example, if the $R_{ck}$ is equal to 30 MPa, the $f_{ck}$, $f_{cm}$, $f_{ctm}$, $f_{cfm}$ and $E_{cm}$ are 24.9 MPa, 32.9 MPa, 2.6 MPa, 3.1 MPa and 31,447 MPa, respectively.

Looking at the performance of abovementioned example of JRCP, the allowable number of the load repetitions ($N_f$) as function of flexural strength of the slab ($\sigma$), that must not be overcame to avoid fatigue damage can be determined by the PCA method [33] by Equations (9)–(11):

$$N_f = 11.737 - 12.077 \left( \frac{\sigma}{f_{cfm}} \right) \text{ when } \frac{\sigma}{f_{cfm}} \geq 0.55 \tag{9}$$

$$N_f = \left( \frac{4.2577}{\sigma/f_{cfm} - 0.4325} \right)^{3.268} \text{ when } 0.45 < \frac{\sigma}{f_{cfm}} < 0.55 \tag{10}$$

$$N_f = \text{unlimited when } \frac{\sigma}{f_{cfm}} \leq 0.45 \tag{11}$$

### 4.3. Prediction of International Roughness Index (IRI)

IRI is a widely recognized standard used for roughness measurements because of its stability over time and large range of application all around the world. IRI can be predicted according to the formulation proposed by the MEPDG Guide [34], shown in Equation (12):

$$IRI = IRI_i + C_1 P_0 + C_2 SF \tag{12}$$

where $IRI_i$ is the initial value of IRI after the construction (assumed to be 2.5 mm/m [31]), $C_1$ and $C_2$ are experimental coefficients equal to 3.15 and 28.38, respectively, PO is the

number of medium- and high-severity pounchouts (m) determined by Equation (13), and SF is the site shift factor determined by Equation (14),

$$PO = \frac{A_{PO}}{1 + \alpha_{PO}DI_{PO}^{\beta_{PO}}}$$ (13)

$$SF = AGE(1 + 0556FI)(1 + P_{200})10^{-6}$$ (14)

where $A_{PO}$, $\alpha_{PO}$ and $\beta_{Po}$ are calibrated constants equal to 195.789, 19.8947 and $-0.526316$, respectively, $DI_{PO}$ is the accumulated fatigue damage (due to slab bending in the transverse direction) at the end of $y^{th}$ year, AGE is the pavement age (year), FI is the freezing index (°F days) and $P_{200}$ is the percent of subgrade material passing through a No. 200 sieve.

In conclusion, the corrected index $ECR_c$ can cover both the aspects associated with the functionality of the rigid pavement, through a fatigue-related parameter, and the aspects related to the surface characteristics, via the IRI value.

## 5. Conclusions

According to the evidence presented in this paper, it is possible to conclude that the factors provided by SimaPro version 9.5.0.0 or other types of specific LCA software can be exploited to obtain a well-founded and shareable estimation of the EAR index. The introduction of a new index, the ECR, has allowed us also to characterize rigid pavement via an evaluation that includes a functional factor through a fatigue-related parameter and the surface characteristics via experimental curve related to the IRI value.

Building on data on the EPDs available for both asphalt and concrete materials, the normalized EAR values show a variability between 6 and 10 for the mixtures analyzed in this paper. For the concrete, the ECR values yielded for the investigated mixes range from 0.0483 for the Falconara 1 concrete to 0.0955 for the Civitavecchia 3 concrete.

Despite its strong applicability in the pavement sector, the strength of the proposed method lies in the possibility of fine-tuning it to different fields by varying the associated performance coefficients.

In future developments, efforts should be focused on the extension of the methodology to all pavement typologies, and on the definition of additional performance coefficients for rigid pavements that can characterize noise and other aspects, such as the top-down cracking phenomenon.

**Author Contributions:** Conceptualization, all authors; methodology, all authors; software, M.P.; validation, all authors; formal analysis, all authors; investigation, all authors; resources, all authors; data curation, all authors; writing—original draft preparation, all authors; writing—review and editing, all authors. All authors have read and agreed to the published version of the manuscript.

**Funding:** This research received no external funding.

**Data Availability Statement:** The data presented in this study are available on request from the corresponding author. The data are not publicly available due to privacy restrictions.

**Acknowledgments:** The authors would like to thank the AMPLIA and ASPI groups for their support in retrieving data for the research.

**Conflicts of Interest:** The authors declare no conflicts of interest.

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
