# Peer review of "Improving the EAR Index for Flexible Pavement and a Preliminary Definition of an Environmental Index (ECR) for Rigid Pavement"

_constrmater, doi:10.3390/constrmater4010007_

Round 1

Reviewer 1 Report

Comments and Suggestions for Authors

The article contains interesting considerations, and in order to increase its scientific value, it is proposed to include the following comments:
1.    On line 199, list the details of phases A1 to A3 in brackets as listed on line 208 for A4 and A5.
2.    Figure 2 – please equalise the number of decimal places in the EAR values.
3.    Figure 4 - please graphically standardise the names of concrete mixtures (as in Figure 2, the names of asphalt mixtures), the number of decimal places. Instead of EAR, there should be ECR.
4.    Has a comparative analysis of the mentioned concrete mixtures been carried out, taking into account the mechanical properties (ECRc), using samples from experimental tests?

5.    Please correct the list of references in accordance with the guidelines for Authors.

Author Response

The authors agree with the review and the paper was amended according to the comments. For the suggestion 4, the comparative analysis of different cement mixtures will be carried out in the next step of the research

Reviewer 2 Report

Comments and Suggestions for Authors

The article is very topical, as a very topical issue of EPD evaluation is examined and analyzed. The article is also innovative in that it offers an improved approach to perform an environmental assessment.

Some notes:

1. Typo on line 121 - "EOL(910)"

2. In order to give these articles an even greater scientific value as an applied research, it would be desirable to include in the references several scientific articles related to this topic, for example: DOI : 10.3390/su132212487, because quite a lot of technical references (EPD certificates), which are good, but should be increased amount of scientific papers.

3. The conclusions are too general, it is desirable to supplement them with numerical values, as the article analyzed several data from real factories

Author Response

The paper was revised adding new scientific articles to the references and modifying the conclusions with numerical evidences

Reviewer 3 Report

Comments and Suggestions for Authors

The article describes a novel method for analyzing pavement materials, notably asphalt and cement-bound mixtures, with an emphasis on both technical and economic needs, as well as increased environmental performance. The methodology expands on a prior study that offered a reference parameter for evaluating technical offers and assigning points based on environmental effect during the tender phase.

The paper revises the Environmental Asphalt Rating methodology, emphasizing main variations and improvements related to the update of the ISO standard. Additionally, it introduces the Environmental Concrete Rating (ECR) for rigid pavement, following a similar approach as the Environmental Asphalt Rating.

The suggested method's strengths include its ability to integrate environmental and mechanical performance for asphalt and cementitious concrete pavements. It considers functionality through a fatigue-related parameter and surface features related to the International Roughness Index (IRI) value while correcting the ECR index for cementitious concrete pavements. The essay emphasizes the method's flexibility to diverse areas by adjusting associated performance coefficients.

However, without specific details about the actual content of the Environmental Asphalt Rating methodology and the Environmental Concrete Rating, it is challenging to provide a thorough evaluation of the article's contribution. The effectiveness of the proposed method would depend on the accuracy and relevance of the introduced parameters, as well as the practical applicability of the methodology in real-world road infrastructure projects. Additionally, the article's impact would be influenced by the extent to which it addresses current challenges in pavement construction and aligns with industry standards and practices.

The paper's conclusions underline the feasibility of applying SimaPro or other particular Life Cycle Assessment (LCA) software elements to produce a well-founded and shared assessment of the Environmental Asphalt Rating (EAR) index. The introduction of the Environmental Concrete Rating (ECR) is regarded as a significant step because it enables the characterization of rigid pavement by incorporating a functional factor via a fatigue-related parameter and surface characteristics via an experimental curve related to the International Roughness Index (IRI) value.

While the conclusions of the paper highlight the potential benefits and strengths of the proposed methodology, it's important to consider potential weaknesses as well.

Pay attention to:

·         The report mentions the "strong correlation with the pavement sector," but it does not address the application of the proposed methodology to new settings or geographic regions.

·         The efficiency of the system may vary depending on environmental circumstances, material availability, and construction practices.

·         The study makes no mention of potential difficulties or restrictions in applying the proposed methodology in real-world construction projects.

·         While the report discusses the opportunity to fine-tune the process by modifying associated performance coefficients, it does not go into the difficulties or considerations involved in choosing or adjusting these coefficients. The lack of specifics may cause readers to doubt the methodology's applicability.

To summarize, while the paper proposes a promising methodology for environmentally conscious pavement characterisation, these possible shortcomings should be explored and solved in order to improve the robustness and application of the suggested strategy.

Author Response

The Authors wish to thank the review for the suggestions received and the very interesting point of view. All the topics discussed will be included in the following steps of this research
